# Simplified TAVR Procedure: How Far Is It Possible to Go?

**DOI:** 10.3390/jcm11102793

**Published:** 2022-05-16

**Authors:** Florence Leclercq, Pierre Alain Meunier, Thomas Gandet, Jean-Christophe Macia, Delphine Delseny, Philippe Gaudard, Marc Mourad, Laurent Schmutz, Pierre Robert, François Roubille, Guillaume Cayla, Mariama Akodad

**Affiliations:** 1Department of Cardiology, Montpellier University Hospital, 34295 Montpellier, France; pierrealain.meunier@gmail.com (P.A.M.); jc-macia@chu-montpellier.fr (J.-C.M.); d-delseny@chu-montpellier.fr (D.D.); francois.roubille@gmail.com (F.R.); akodadmyriam@gmail.com (M.A.); 2Department of Anesthesiology and Critical Care Medicine, Montpellier University Hospital, 34295 Montpellier, France; t-gandet@chu-montpellier.fr (T.G.); p-gaudard@chu-montpellier.fr (P.G.); m-mourad@chu-montpellier.fr (M.M.); 3Department of Cardiology, CHU Nimes, Nimes University Hopstal, 30029 Nimes, France; laurent.schmutz@chu-nimes.fr (L.S.); pierrecardio@gmail.com (P.R.); cayla.guillaume@gmail.com (G.C.); 4PhyMedExp, INSERM U1046, CNRS UMR 9214, Montpellier University Hospital, 34090 Montpellier, France

**Keywords:** TAVR, simplification, tailored approach

## Abstract

Increasing operators’ experience and improvement of the technique have resulted in a drastic reduction in complications following transcatheter aortic valve replacement (TAVR) in patients with lower surgical risk. In parallel, the procedure was considerably simplified, with a routine default approach including local anesthesia in the catheterization laboratory, percutaneous femoral approach, radial artery as the secondary access, prosthesis implantation without predilatation, left ventricle wire pacing and early discharge. Thus, the “simplified” TAVR adopted in most centers nowadays is a real revolution of the technique. However, simplified TAVR must be accompanied upstream by a rigorous selection of patients who can benefit from a minimalist procedure in order to guarantee its safety. The minimalist strategy must not become dogmatic and careful pre-, per- and post-procedural evaluation of patients with well-defined protocols guarantee optimal care following TAVR. This review aims to evaluate the benefits and limits of the simplified TAVR procedure in a current and future vision.

## 1. Introduction

The care pathway for patients undergoing transcatheter aortic valve replacement (TAVR) was initially based on open surgery standards and often included general anesthesia (GA), per-procedural invasive monitoring, systematic transesophageal echocardiography and intensive care unit (ICU) admission. Considering improvements in procedural outcomes and the decreasing surgical risk of patients undergoing TAVR, fast-track protocols were recently developed for transfemoral TAVR in numerous centers and tend now to generalize [1,2,3,4,5,6,7]. The aims of simplifying the procedure were mainly to allow rapid recovery and early ambulation and discharge and to improve patient satisfaction and to reduce hospital costs without compromising safety.

While simplification does not compromise success and safety, identifying potential candidates for a simplified strategy during TAVR requires rigorous pre-, per- and post-procedural patient’s clinical evaluation.

## 2. Sedation or General Anesthesia

The less invasive and simplified transfemoral TAVR procedure, including percutaneous access and the absence of systematic transesophageal echocardiography (TEE), allows performing the procedure under local anesthesia with or without conscious sedation (LA-CS). However, evidence supporting the choice of whether to conduct a TAVR procedure under GA or LA-CS among patients eligible for both approaches is provided mainly by non-randomized trials and registry data [8,9]. The SOLVE TAVI trial is the only randomized study comparing GA to LA-CS. This multicenter trial did not show any significant difference between the two techniques regarding a 30-day combined primary safety endpoint [10]. An American multicenter TAVR registry including 120,080 patients from 2016 to 2019 showed that LA-CS was associated with a small but significant decrease in in-hospital mortality (0.2%; *p* = 0.01) and 30-day mortality (0.5%; *p* < 0.001) [11]. Conversely, in the meta-analysis of Husser et al., including 16,543 patients from 2011 to 2014, no significant difference in 1-year mortality was highlighted between both techniques without significant difference in myocardial infarction, acute renal failure, and pacemaker implantation [12]. A recent TVT registry showed a trend toward conscious sedation for TAVR with an increase in TAVR operator experience [13].

To date, no clear recommendation is provided for the choice of anesthetic techniques during TAVR procedures [14]. GA is usually preferred for the non-femoral approaches, particularly for the subclavian or the carotid access, which are more painful for the patient. GA would also provide the practitioner and the patient more comfort when the procedure is anticipated to be long and complex. Moreover, anxious patients or a language barrier can make LA-CS both ineffective and difficult to assess. Furthermore, the use of GA during the TAVR procedure does not necessarily require the use of associated invasive interventions, such as arterial or urinary catheters, which can cause complications and increase the length of the procedure. A minimalist procedure is thereby possible in selected patients when the GA strategy is selected [15]. LA with or without conscious sedation is increasingly used and is usually preferred in the youngest patients without major comorbidity. The overall simplification of the procedure included in the minimalist strategy probably contributes more to the better outcomes than to the anesthesia technique itself.

The optimal anesthetic strategy has probably to be determined upstream of the procedure by the Heart team, similarly to how vascular access or prosthesis size are selected. Conversion from LA-CS to GA occurred in 5.9% of patients in the SOLVE TAVI trial and in 7.9% of patients in the meta-analysis of Villablanca et al., where cross-over to GA was associated with higher mortality [10,16]. Although complications may not be all anticipated, the best choice of anesthesia strategy regarding patient profile may avoid anesthetic conversion related to comorbidities or complex anatomies.

### Hybrid Room or Catheterization Laboratory

Hybrid operating rooms have been considered the ideal location to perform TAVR. They combine features of both the catheterization laboratory (high-quality fluoroscopy and imaging) and surgical facilities with high cleanliness level and laminar airflow and positive pressure. Hybrid rooms allow prompt surgical conversion in the case of severe complications (e.g., annular rupture or any “bail-out” options, including peripheral intervention and/or need for mechanical support). However, hybrid rooms are not available in all centers performing TAVR. In the study of Spaziano et al. with 12,121 patients from the FRANCE TAVI registry, the authors showed that the midterm outcomes were similar in either hybrid room or catheterization laboratory. Moreover, in this study, hybrid room location was paradoxically associated with an increase in bleeding and suspected infection requiring antibiotics. The overall rates of procedural complications were however low in both groups [17]. These results may be probably partly explained by higher-risk patients—those with alternative access, high-risk coronary or vascular anatomy, bicuspid valves or other anatomical issues—who underwent TAVR in the hybrid room.

With the increasing use of a minimalist approach, a high number of TAVR procedures are performed in the catheterization laboratory. Although a simplified TAVR approach in the catheterization laboratory appears as a safe strategy, hybrid rooms, when available, can be proposed for selected high-risk patients.

## 3. Vascular Access

### 3.1. Percutaneous vs. Surgical Femoral Approach

A safe vascular approach is the cornerstone of TAVR procedure success as a vascular complication (VC) remains one of the main issues. Over the years, a significant reduction in major VC after TAVR was observed, occurring in less than 5% of low-risk patients in more recent trials and registries [4,5,18].

The percutaneous (PC) approach was developed in transfemoral (TF) TAVR and progressively replaced surgical cutdown (SC) performed in the initial experience. Indeed, the PC approach is considered less invasive, especially with the use of smaller-diameter sheaths and ultrasound-guided puncture, which may be associated with a shorter hospitalization length compared to the SC approach [19,20,21,22]. LA may also be facilitated in the PC approach without necessarily requiring the presence of the surgeon in the catheterization laboratory [21]. However, SC may be considered more accurate for vessel puncture into calcified vessels, and vascular control may be better in the case of adverse events, particularly in obese patients regarding the long skin to the artery path [23]. Moreover, by using the surgical preclose technique to avoid arterial cross-clamping and pursing effect, favorable results were previously reported with a low rate of VC [23,24].

A tailored approach has been proposed by Olasinska et al. and has shown a five-fold reduction in VC in a study considering SC for TF TAVR in the case of “vascular risk findings for PC” in computed tomography (CT), including calcifications, diffuse atherosclerosis, small diameters or tortuosity [25].

The PC approach, widely used nowadays, should be the default strategy for TF TAVR to make the procedure as “minimalist” as possible. However, the SC approach may be considered in selected high-risk profile patients as associated with these patients with a low incidence of VC and bleeding in experienced hands.

### 3.2. Femoral vs. Alternative Access

European guidelines recommend the transfemoral (TF) approach for first-line access during transcatheter aortic valve replacement (TAVR) when the vessel anatomy is favorable [14]. Indeed, although evidence for transfemoral TAVR from randomized trials is robust, observational data are primarily available for alternative access TAVR. However, some anatomic challenges, such as low vessel diameter, vascular tortuosity, calcifications or aortoiliac vascular pathology, may increase the risk of procedural complications and failure, leading to the prospect of alternative access sites [26,27]. Transaortic and transapical accesses are invasive hybrid approaches and are associated with less favorable results than the TF approach [28,29]. Subclavian access may be difficult in the case of obesity, small artery diameter or important calcifications and may be avoided in the case of patent mammary artery bypass [30]. The transcarotid (TC) approach, initially developed by Modine et al. in 2010 [27], showed encouraging results in terms of feasibility and safety [31,32].

In the case of complex femoral anatomy (defined in CT scans as iliofemoral diameters between 5.5–6 mm or <6.5 mm with severe calcifications or tortuosity and/or abdominal aorta pathology), favorable results were reported with the TC approach, with a decrease in major bleeding complications despite the higher risk profile of patients [33].

Direct access from the sheath to the ascending aorta provides good control over the valve positioning and avoids multiple manipulations in the aortic arch. The TC approach should however be performed by an experienced vascular surgeon due to the presence of important local structures, such as the vagus nerve and the respiratory system. This approach should therefore be considered a good alternative in the case of challenging but feasible vascular anatomy in patients undergoing TAVR.

### 3.3. Minimalist Secondary Access

#### Radial Access

Secondary vascular access is required for angiographic guidance during prosthesis deployment. Although vascular complications dramatically decreased in TF TAVR, one-fourth of these complications occurred at the secondary femoral access site, inciting the operators to use the radial artery for secondary access [34]. Radial access has been adopted as the default strategy in numerous centers and may decrease all vascular complications following TAVR, improve patient comfort and allow earlier ambulation [34].

However, radial access may be particularly challenging on the right in the case of tortuous vessels or spasms. Furthermore, managing the main femoral access vascular complication from the radial access may be challenging but feasible with appropriate tools [34].

Then, although the radial artery may be favored for secondary arterial access, secondary femoral access may be required in the case of failure or an emergent need of cross-over. The use of a hydrophilic sheath coating to reduce the incidence of radial artery spasms may also be promoted [35]. Distal radial access at the anatomical snuffbox was recently proposed for secondary access in TAVR and seems to be superior to proximal access in preventing radial occlusion. However, larger randomized trials are needed to further evaluate the advantages of distal over proximal radial access in TAVR [36].

## 4. Left Ventricle Wire Pacing

Intraprocedural rapid ventricular pacing is still necessary to ensure a transient cardiac standstill during the deployment of a balloon-expandable transcatheter heart valve (THV) and pre- and post-dilatation when required. A transvenous temporary pacing lead is traditionally performed at the right ventricular (RV) apex and requires additional venous access, usually femoral, with the inherent risk of vascular complications, bleeding or infection. There is also a risk of RV perforation and cardiac tamponade, particularly in elderly patients [37]. Similar to the technique of LV stimulation used in pediatric valvuloplasty, an LV pacing technique using the valve delivery guidewire was evaluated by Faurie et al. [38]. In the EASY TAVI study, the authors showed that, compared with RV stimulation, LV stimulation provided via the 0.035-inch stiff guidewire during TAVR was associated with significant reductions in procedure duration, fluoroscopy time and cost, with similar efficacy and safety. This strategy has to be favored as often as possible, but preventive femoral vein puncture may be required for some patients in the presence of high-degree conduction disturbance before TAVR (bifascicular block) for rapid temporary pacing after catheter guiding and LV wire removal. Finally, a temporary external pacemaker may represent a safer alternative to femoral temporary lead in patients with a high risk of conduction disorders and may allow early mobilization [39].

## 5. Direct Implantation of the THV

New-generation THVs are associated with low-profile and orientable delivery systems that facilitate valve crossing. These systems have been associated with high TAVR success rates without prior dilatation of the native valve in observational studies and registries. Two recent randomized studies have shown that direct implantation of the THV was non-inferior to the conventional procedure using systematic balloon valvuloplasty according to device success as per the valve academic research consortium (VARC)-2 [40] and in procedural adverse events, particularly aortic regurgitation and pacemaker rate. Procedural times, contrast volume and radiation doses were not statistically different between the two strategies [41,42].

Although the absence of predilatation did not increase the post-dilatation rate in the DIRECTAVI trial using balloon-expandable THV, the post-dilatation rate was reported with a two-fold increase in the direct trial following the direct implantation of a self-expandable THV [41]. It might also be necessary to use pre-dilatation to cross the valve in patients with very tight stenosis with greater degrees of calcification, as reported in the recent EASE-IT TF registry and in the DIRECTAVI trial [42,43]. Other anatomic factors, including calcified bicuspid valves, horizontal arch and/or femoral tortuosity, may indicate crossing difficulties in the absence of pre-dilatation [42].

Thus, as the direct implantation of the THV may be used as the default strategy in most patients, particularly with new-generation balloon expandable valves, careful evaluation of valvular and aortic anatomy on the CT scan is required to anticipate difficulties of crossing the valve and to select patients requiring aortic valve predilatation.

## 6. Per-Procedural Echography

Considering the high immediate success rate of TAVR, the minimalist strategy does not include systematic echocardiography to assess the result of the procedure in the catheterization laboratory [6,7]. However, an experienced echocardiographer must be available during each TAVI procedure. Echocardiography is needed to evaluate any potential complications and establish the degree of residual aortic regurgitation. Transthoracic echocardiography (TTE) is the default exam, while TEE is only performed when the TTE window is not adequate or if a specific issue needs to be evaluated. TEE can be performed for a short time interval, and only light sedation is needed. In some studies, intraprocedural TEE was associated with a lower incidence of paravalvular leakage [44].

Considering the extension of TAVR indications to lower risk and younger patients, optimal results of the procedure without mismatch and without aortic regurgitation is a real issue to improve patients’ prognosis and THV durability [45]. TTE and/or TEE should be considered if any doubt about the optimal results of the procedure to evaluate precisely the THV position and function and consider possible alternative options (post-dilatation, redo TAVR, etc.).

## 7. Short or No Monitoring

Telemetry monitoring (TM) with or without ICU admission is usually considered the standard of care after TAVR [14].

Selecting patients who need an electrocardiogram (ECG) monitoring according to the risk of conductive disturbances was pointed out by Toggweiller et al. and described by our team [15,46].

A strategy of selective TM after TAVR according to the risk of adverse events, particularly the risk of conductive disorders, may be proposed to limit TM units and ICU overload in high-volume TAVR centers, allowing the admission of low-risk patients in the general cardiology ward without TM. With a rigorous selection of patients, TAVR can be routinely and safely performed without systematic TM and ICU admission in at least one-third of patients [15]. Atrial pacing post-TAVR was recently proposed to identify patients who may benefit from extended rhythm monitoring [47].

To date, there are no clear guidelines or recommendations for pacemaker implantation post TAVR except in patients with complete heart block and high-grade AV block persisting more than 7 days post-THV implantation [14]. However, expert opinions have been recently provided, especially regarding new left bundle branch block management [48].

With the decrease in hospitalization length after TAVR, the risk is to implant pacemakers in a lower degree of conductive disorders in order to decrease ICU or monitoring care. Longer monitoring is probably required for these patients to limit unnecessary pacemaker implantation if the conductive disorder is stable or regressive. Specific studies regarding this major issue are required.

## 8. Early Discharge

There is an increasing trend toward shorter hospital stays after TAVR procedures, in particular for patients undergoing the procedure via transfemoral access [1,6,7].

Despite improvements in the results, TAVR remains associated with specific complications, primarily vascular complications and conductive disorders, which can increase the hospitalization length of stay [6,7].

In the FRANCE TAVI registry, variables associated with late discharge were female sex, co-morbidities, major complications, self-expandable valve, general anesthesia and a significant center effect [49].

The FAST-TAVI registry validated the appropriateness of a pre-specified set of risk criteria that allow a safe and timely discharge. The rate of 30-day complications did not reveal any risk increase with this strategy compared with the reported outcomes in major TAVR trials and registries [6]. However, logistic or social reasons remained the main barriers to early discharge, particularly in older and frail patients. Furthermore, considering the risk of some conductive disorders worsening, the objective of a shorter hospitalization length must not be obtained at the expense of safety. Thus, carefully selecting patients who may benefit from an early discharge strategy is the key. Algorithms have been recently proposed to predict patients’ eligibility for early discharge, and this strategy has demonstrated its safety and efficacy [50].

## 9. Conclusions

Both improving the results of the procedure and extending indication to lower-risk patients explained the global simplification of TAVR regarding the technique itself, post-procedural monitoring or early discharge.

This streamlined journey has not become simplistic, and careful evaluation of each patient remains mandatory to propose a TAVR upstream strategy according to individual patient characteristics (Figure 1).

## Figures and Tables

**Figure 1 jcm-11-02793-f001:**
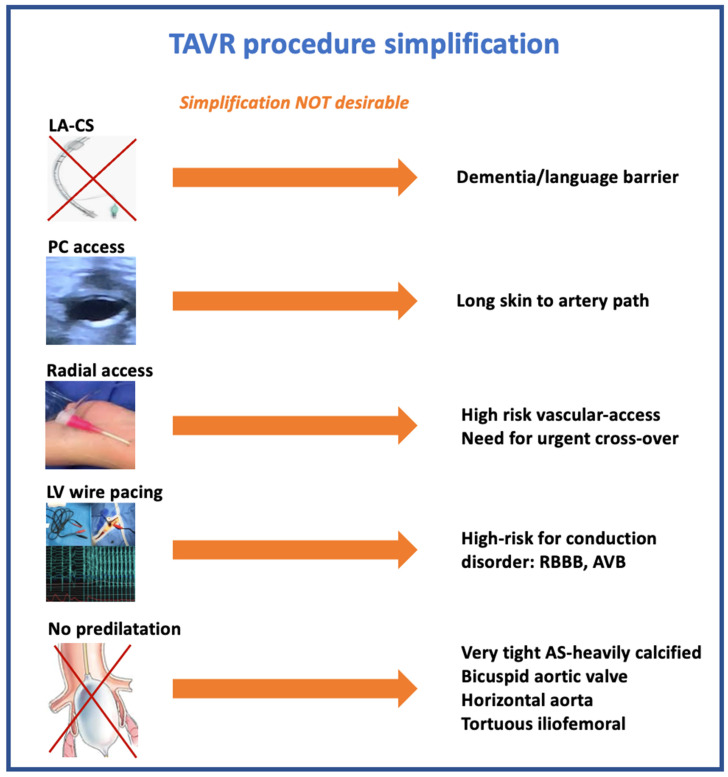
TAVI procedure simplification strategy: desirable or not.

## Data Availability

This study reported any data but only studies.

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
