# Peer review of "Simplified TAVR Procedure: How Far Is It Possible to Go?"

_jcm, 2022, doi:10.3390/jcm11102793_

Round 1

Reviewer 1 Report

I read with interest review article by Leclerq et al titled: “Simplified TAVR procedure: how far is it possible to go?”

In the introduction I would like to suggest to the authors to cite a recent paper form JACC on same day discharge, ”The Multicenter PROTECT TAVR Study”

https://www.jacc.org/doi/abs/10.1016/j.jcin.2021.12.046

Under the heading of sedation or general anesthesia, I suggest that the author add data form the recent TVT registry study showing that the trend of conscious sedation for TAVR has increases with the increase TAVR operator experience.

Under heading 7, short or no monitoring, authors can consider discussion the use of Rapid Atrial Pacing to predict PPM implantation post TAVR.

https://www.jacc.org/doi/abs/10.1016/j.jcin.2020.01.215

Author Response

Reviewer #1

I read with interest review article by Leclerq et al titled: “Simplified TAVR procedure: how far is it possible to go?”

1/ In the introduction I would like to suggest to the authors to cite a recent paper form JACC on same day discharge, ”The Multicenter PROTECT TAVR Study”

 https://www.jacc.org/doi/abs/10.1016/j.jcin.2021.12.046

Answer:   We cited the recent paper published in JACC intervention on same day discharge (reference 7)

2/ Under the heading of sedation or general anesthesia, I suggest that the author add data form the recent TVT registry study showing that the trend of conscious sedation for TAVR has increases with the increase TAVR operator experience.

Answer:   Data from the TVT registry study was added in the heading "sedation or general anesthesia": “Recent TVT registry study showed that the trend of conscious sedation for TAVR has increases with the increase TAVR operator experience (13)” (page 2, line 63-64) and a new reference was cited (13)

3/ Under heading 7, short or no monitoring, authors can consider discussion the use of Rapid Atrial Pacing to predict PPM implantation post TAVR.

https://www.jacc.org/doi/abs/10.1016/j.jcin.2020.01.215

Answer:   Under heading 7 "short or no monitoring" we discussed the use of rapid atrial pacing to predict PPM implantation post TAVR: “Atrial pacing post-TAVR was recently proposed to identify patients who may benefit from extended rhythm monitoring (45)” (page 6, line 479-480):  and reference 47 was added.

Reviewer 2 Report

In this interesting review, Leclercq et al. summarized the multiple steps taken over time to simplify TAVR procedures

Comments :

1) The manuscript would be greatly improved by any table or figure summarizing every step taken to simplify TAVR. It could, for example, illustrate situations where simplifying might not be recommended (e.g. complex femoral anatomy or RBBB…)

2) TAVR from alternative access does not have the robust body of evidence gathered by TF TAVR (albeit a lot of observational data). It might be prudent to mention it.

3) You mention that crossover might be harder from secondary radial access compared to femoral. Do you have evidence? Sure it requires specific equipment, but it is manageable.

4) You could mention temporary external pacemaker as an alternative to femoral venous temporary lead (especially in patients with RBBB or other high-risk conduction disturbances). It provides safe protection with time to evaluate the need for a permanent pacemaker while allowing patients to be mobilised (and it prevents complications, as seen in the MobiTAVI trial). If you need an idea of what I’m talking about : J Innov Card Rhythm Manag. 2019 May 15;10(5):3652-3661.

5) There are no guidelines on indications for permanent pacemaker implantation after TAVR but there are expert opinions. You might want to cite them.

Minor comments:

1) Please add a citation justifying the ¼ of vascular complications for secondary access.

2) Please define and cite VARC-2 criteria

3) Abbreviation needs to be defined when first used. (TTE, TEE, THV)

4) I don’t quite get the sentence line 185 – could you rephrase?

Author Response

Reviewer #2

In this interesting review, Leclercq et al. summarized the multiple steps taken over time to simplify TAVR procedures

1/ The manuscript would be greatly improved by any table or figure summarizing every step taken to simplify TAVR. It could, for example, illustrate situations where simplifying might not be recommended (e.g. complex femoral anatomy or RBBB…)

Answer:   Thank you for your valuable comment. We modified the figure 1 and added situations where simplification may not be desirable for each step.

2/ TAVR from alternative access does not have the robust body of evidence gathered by TF TAVR (albeit a lot of observational data). It might be prudent to mention it.

Answer:   Thank you for this comment. We added this in the appropriate section: “Indeed, although evidence for transfemoral TAVR from randomized trials are robust, mainly observational data are available for alternative access TAVR” (page 4, line 185-186).

3/ You mention that crossover might be harder from secondary radial access compared to femoral. Do you have evidence? Sure it requires specific equipment, but it is manageable.

Answer:   We would like to thank the reviewer for this comment. We mentioned this point in the appropriate section “Furthermore, managing main femoral access vascular complication from the radial access may be challenging but feasible with appropriate tools [34].” (page 4, line 215).

4/ You could mention temporary external pacemaker as an alternative to femoral venous temporary lead (especially in patients with RBBB or other high-risk conduction disturbances). It provides safe protection with time to evaluate the need for a permanent pacemaker while allowing patients to be mobilised (and it prevents complications, as seen in the MobiTAVI trial). If you need an idea of what I’m talking about : J Innov Card Rhythm Manag. 2019 May 15;10(5):3652-3661.

Answer:   thank you for this comment. We mentioned this point in the appropriate section: “Finally, temporary external pacemaker may represent a safer alternative to femoral temporary lead in patients with high risk of conduction disorder and may allow early mobilization” (page 5, line 272-274).

5/ There are no guidelines on indications for permanent pacemaker implantation after TAVR but there are expert opinions. You might want to cite them.

Answer:   Thank you for your comment, reference 46 has been added.

6/ Please add a citation justifying the ¼ of vascular complications for secondary access.

Answer:  thank you for this comment. Reference 34 was cited to support this data.

7/ Please define and cite VARC-2 criteria

Answer:  thank you for this comment. We defined and cited the reference for VARC-2 criteria (40).

8/ Abbreviation needs to be defined when first used. (TTE, TEE, THV)

Answer:  corrected.

9/ I don’t quite get the sentence line 185 – could you rephrase?

Answer:  thank you for this comment. This was rephrased: “Although the absence of predilatation did not increase post-dilatation rate in the DIRECTAVI trial using balloon-expandable THV, post-dilatation rate was reported with a 2-fold increase in the direct trial following direct implantation of self-expandable THV [41].” (page 5, line 286-288).

Round 2

Reviewer 2 Report

I could not see the attached figure (and it is not mentioned in the text if I'm correct). Nonetheless, all comments were addressed. Great work